# Expression and Biochemical Characterization of a Novel Fucoidanase from *Flavobacterium*
*algicola* with the Principal Product of Fucoidan-Derived Disaccharide

**DOI:** 10.3390/foods11071025

**Published:** 2022-04-01

**Authors:** Yanjun Qiu, Hong Jiang, Yueyang Dong, Yongzhen Wang, Hamed I. Hamouda, Mohamed A. Balah, Xiangzhao Mao

**Affiliations:** 1College of Food Science and Engineering, Ocean University of China, Qingdao 266003, China; 17863152109@163.com (Y.Q.); dongyuey1021@163.com (Y.D.); wangyongzhen204@163.com (Y.W.); hamedhamouda@ouc.edu.cn (H.I.H.); xzhmao@ouc.edu.cn (X.M.); 2Processes Design and Development Department, Egyptian Petroleum Research Institute, Cairo 11727, Egypt; 3Plant Protection Department, Desert Research Center, Cairo 11753, Egypt; mbaziz1974@gmail.com; 4Laboratory for Marine Drugs and Bioproducts of Qingdao National Laboratory for Marine Science and Technology, Qingdao 266237, China

**Keywords:** fucoidan, fucoidanase, GH107 family, cloning and expression, biochemical characterization

## Abstract

Fucoidan is one of the main polysaccharides of brown algae and echinoderm, which has nutritional and pharmacological functions. Due to the low molecular weight and exposure of more sulfate groups, oligo-fucoidan or fucoidan oligosaccharides have potential for broader applications. In this research, a novel endo-α-1,4-L-fucoidanase OUC-FaFcn1 which can degrade fucoidan into oligo-fucoidan was discovered from the fucoidan-digesting strain *Flavobacterium algicola* 12,076. OUC-FaFcn1 belongs to glycoside hydrolases (GH) family 107 and shows highest activity at 40 °C and pH 9.0. It can degrade the α-1,4 glycosidic bond, instead of α-1,3 glycosidic bond, of the fucoidan with a random tangent way to generate the principal product of disaccharide, which accounts for 49.4% of the total products. Therefore, OUC-FaFcn1 is a promising bio-catalyst for the preparation of fucoidan-derived disaccharide. These results further enrich the resource library of fucoidanase and provide the basis for the directional preparation of fucoidan-derived oligosaccharide with specific polymerization.

## 1. Introduction

Fucoidan, a sulfated and fucose-rich polysaccharide that is isolated from brown alga and echinoderm, was first extracted from *Ascophyllum nodosum*, *Fucus vesiculosus*, and *Laminaria digitata* by Kylin in 1913 [1]. Afterwards, researches proved that sulfated fucans are also present in sea cucumbers (*Ludwigothurea grisea*) [2] and seagrasses (*Halodule pinifolia*) [3]. Fucoidans from marine echinoderms possess the regular linear structure, and have a simpler structure than algae-derived fucoidans [4]. Similar to other seaweed polysaccharide, fucoidan exhibits a series of biological activities, such as anticancer, antioxidant, anti-inflammatory, tuning immune-responses anticoagulant, antiangiogenic, anti-infectious, and anti-pathogen [5,6]. 

Both the structure and composition of fucoidan are related to algae species, harvest time, and extraction methods. For example, it has been reported that fucoidans are more abundant in the reproductive stage [7,8]. Fucoidan not only has the main chain that is composed of fucose, xylofucoglucuromannans, fucoglucuronans, and fucogalactans, but also has side chain that is made up of xylose, galactose, mannose, and rhamnose [9]. In general, fucoidan that is composed of fucose is the most commonly discussed and its backbone structure can be divided into two groups [10]. The fucoidans that are extracted from the algae *Laminaria digitata*, *L. saccharina*, *Cladosiphon okamuranus*, and *Chorda filum*, as well as sea cucumber have the backbone consisting of (1 → 3)-linked-L-fucopyranose residues with sulphate groups that are located in C4 or C2 position of L-fucopyranose. The second group, including fucoidans from *Ascophyllum nodosum* and *Fucales* genus, have the backbone structure consisting of repeating (1 → 3)- and (1 → 4)-linked-L-fucopyranose residues, whereas the sulphation occurs in C2, C3, C4, or sometimes C6 of L-fucopyranose with branch chains in C2 [11]. The chemical structures of fucoidans are complex and vary with species and seasonal variations, which all caused the low utilization of fucoidan. Silchenko et al. [12] proved that 4O-sulfation makes a significant contribution to the anticancer activity of fucoidans. Compared to fucoidans, oligo-fucoidans exhibit higher biological activities and nutritional functions due to their low molecular weight (MW) and exposure of more sulfate groups [12,13,14,15,16]. Moreover, the degree of polymerization has an important effect on the biological activity and nutritional function of oligo-fucoidans. For instance, Liu et al. [14] found that the high MW fucoidans from *L. japonica* show no antibacterial activity, but their depolymerized products (<6 kDa) with abundant sulfate groups can effectively inhibit the proliferation of bacteria and have the antibacterial activity. A review [16] also concluded that the fucoidans with low MW (<10 kDa) and high sulfation display higher anti-diabetic activity. Therefore, it is of great significance to transform these irregular macromolecular polysaccharides to regular and well-characterized oligo-fucoidans with broad research and practical applications. Some effective degradation methods should be established to prepare low molecular weight fucoidan. In general, approaches for polysaccharide degradation includes physical degradation, chemical degradation, and enzymatic degradation. Physical degradation includes ultrasonic, microwave, plasma treatment, and subcritical hydrolysis. Chemical degradation includes acid/alkali hydrolysis and oxidative degradation. Among them, specific enzymatic preparation is the most gentle and efficient approach for the production of oligo-fucoidan with specific structure.

Many studies have found that some marine bacterium could degrade fucoidan [17,18] and the hydrolases that are responsible for fucoidan degradation also were discovered in the intestinal and digestive glands of sea cucumber [19]. However, there are a few reports of fucoidanases in the CAZy database (http://www.cazy.org/Home.html) (accessed on 1 July 2021), which are unified in the glycoside hydrolases families GH107 and GH168. The GH107 family mainly incudes endo-α-1,4-L-fucoidanase (EC 3.2.1.212) which can hydrolyze the fucoidan from algae and 14 enzymes are categorized into this group, including Fda1 and Fda2 from *Alteromonas* sp. SN-1009 [20], MfFcna from *M. fucanivornans* [21], FFA1 [22] and FFA2 [23] from *Formosa algae*, P19DFcnA [24] from *Psychromonas* sp. SW19D, P5AFcnA from *Psychromonas* sp. SW5A, Fhf1 [25] and Fhf2 [26] from *Formosa haliotis*, FWf1 [27] and FWf2 from *Wenyingzhuangia fucanilytica* CZ1127^T^, and three unknown sources of fucoidanases [28]. The GH168 family is a novel class containing only one fucoidanase, named FcnA, which was isolated from *W. fucanilytica* CZ1127^T^ [29]. However, due to the complex structure of fucoidan, degradation efficiency, substrate specificity, and product composition of fucoidanases towards different fucoidan have certain limitations. Therefore, enriching the resource library of fucoidanases is significant for the production of oligo-fucoidan with a specific structure and biological activity. 

Studies have shown that *Flavobacterium algicola* TC2^T^ is a fucoidan-digesting strain and can grow well by using fucoidan as the sole carbon source [30], suggesting that *F. algicola* is a potential producer of enzymes with fucoidan-hydrolyzing activity. In this research, a novel endo-α-1,4-L-fucoidanase OUC-FaFcn1 from *F. algicola* 12,076 was discovered and expressed in the host of *E. coli*. Meanwhile, the catalytic properties of the OUC-FaFcn1 and its principal products during fucoidan hydrolysis were investigated. 

## 2. Materials and Methods

### 2.1. Chemicals, Strains, and Culture Conditions

Fucoidan was purchased from Solarbio (Beijing, China). Fucoidan from *Laminaria japonica* was purchased from Rizhao Jiejing Ocean Biotechnology Development Co., LTD (Rizhao, China). Fucoidan from sea cucumber *Isostichopus badionotus*, *Acaudina molpadioidea,* and *Apostichopus japonicus* were kindly provided by Prof. Yaoguang Chang (College of Food Science and Engineering, Ocean University of China). 4-Hydroxybenzhydrazide (pHBH) and L-fucose were purchased from Macklin (Shanghai, China). Other chemicals were all analytical grade unless otherwise specified.

The *F. algicola* 12,076 strain that was purchased from China General Microbiological Culture Collection Center (CGMCC) was cultured in the medium containing fucoidan as the sole carbon source, which was composed of 0.2% (*w*/*v*) fucoidan, 1% NaCl, 0.2% K_2_HPO_4_, 0.5% (NH_4_)_2_SO_4_, 0.2% MgSO_4_·7H_2_O, and 0.01% FeSO_4_·7H_2_O. *E. coli* DH5α and *E. coli* BL21 (DE3) that were used for amplify the recombinant plasmid and protein expression were bought from TSINGKE BIOTECH (Qingdao, China) CO., LTD. *E. coli* DH5α was cultivated in LB medium that was composed of 1% (*w*/*v*) tryptone, 1% NaCl, and 0.5% yeast extract, at 37 °C and 220 rpm for 12 h. *E. coli* BL21 (DE3) was cultivated in ZYP-5052 medium consisting of 1% (*w*/*v*) tryptone, 0.5% yeast extract, 0.5% glycerin, 0.05% glucose, 0.2% α-galactose, 0.05% MgSO_4_, 0.7% Na_2_HPO_4_, 0.7% KH_2_PO_4_, and 0.3% (NH_4_)_2_SO_4_, at 20 °C for 48 h.

### 2.2. Gene Cloning and Plasmid Construction

The total genome DNA of *F. algicola* 12,076 was obtained by the Bacterial Genome DNA Extraction Kit (TIANGEN BIOTECH, China). On account of the full genome sequence (Accession No. JAJTUT010000001-JAJTUT010000006) of *F. algicola* 12,076 [31,32], a pair of primers (OUC-FaFcn1F: 5’-CGAATTCGAGCTCCGCAAGTTCCAGATTCCG-3’ and OUC-FaFcn1R: 5’-TCAGTGGTGGTGGTGGTGGTGTTGTTTTATTTTTATG-3’) was designed and synthesized based on the conserved sequences of the genes encoding fucoidanases in *M. fucanivornans* (CAI47003.1), *F. algae* (WP_057784217.1 and WP_057784219.1), *F. haliotis* (WP_066217780.1), and *W. fucanilytica* CZ1127^T^ (ANW96098.1 and ANW96097.1) to clone the putative gene encoding fucoidanase from its genomic DNA. The purified DNA products were inserted into the plasmid pET-28a(+) by the homologous sequences that are marked by an underline in primers and sequenced. The signal peptide sequences were predicted by SignalP-5.0 (http://www.cbs.dtu.dk/services/SignalP/) (accessed on 1 July 2021). The recombinant plasmid pET-28a(+)-OUC-FaFcn1 harboring the fucoidanase OUC-FaFcn1 without signal peptide was constructed and delivered into *E. coli* BL21 (DE3), and then induced foreign enzyme production.

### 2.3. Sequence Analysis and Homology Dynamics Simulations

The phylogenetic tree of the different fucoidanases was constructed by MEGA 7.0. The multi-sequence alignment of fucoidanases was analyzed by ClustalX 2.1 and ESPript 3.0 (http://espript.ibcp.fr/ESPript/ESPript/) (accessed on 1 July 2021). The protein domain was analyzed by InterProScan and the NCBI Domain Database (https://www.ncbi.nlm.nih.gov/Structure/cdd/wrpsb.cgi) (accessed on 1 July 2021). The parameters of OUC-FaFcn1, such as MW and isoelectric point were analyzed by ExPASy ProtParam (https://web.expasy.org/protparam/) (accessed on 1 July 2021). The model of OUC-FaFcn1 was simulated and analyzed by SWISS-MODEL (https://swissmodel.expasy.org/) (accessed on 21 October 2021) and AutoDock 4.2.6 software. The homology modeling structure of OUC-FaFcn1 was obtained by simulating the structures of MfFcnA4 (PDB:6dlh) and MfFcnA9 (PDB:6dns). α-L-fucopyranosyl-2-sulfate was used as the ligand as described by Vickers [24].

### 2.4. Production and Purification of Enzyme

The recombinant *E. coli* BL21(DE3) with pET-28a(+)-OUC-FaFcn1 were cultivated in ZYP-5052 medium at 20 °C, 220 rpm, for 48 h. The cells were harvested by centrifugation (8000× *g*, 5 min at 4 °C), washed three times, and resuspended with 50 mM Tris-HCl buffer (pH 7.6). The collected cells were broken by ultrasonic cell crusher with 360 w power for 20 min, and the supernatant were separated by centrifugation (8000× *g*, 10 min at 4 °C). The crude enzyme solution of OUC-FaFcn1 with His tags were filtered with 0.45 μm filter and then loaded onto a Ni-NTA Sepharose column. OUC-FaFcn1 was obtained by gradient elution with imidazole in different concentrations (20, 50, 100, 200, and 500 mM) in 500 mM NaCl and 50 mM Tris-HCl buffer (pH 8.0). Then, the imidazole was removed from the purified proteins by ultrafiltration with Tris-HCl buffer displacement. The purification effect of OUC-FaFcn1 was analyzed and determined by using SDS-PAGE and Bradford method with bovine serum albumin as the standard [33]. 

### 2.5. Enzyme Activity Assay

Hydrolysis activity of the OUC-FaFcn1 against fucoidan was determined following the method that was described by Shen [29], because pHBH can detect and measure low-level reducing sugar linearly. Specifically, pHBH was dissolved in 2 M HCl with the concentration of 20% (*w*/*v*) and then mixed with 2 M NaOH in a volume ratio of 1:9 [34]. A total of 30 µL purified enzyme solutions were added into 270 µL Tris-HCl buffer (50 mM, pH 9.0) containing 2 mg/mL fucoidan. The reaction mixture was incubated at 35 °C for 30 min and stopped by freezing. Subsequently, the prepared pHBH reagent was added into the reaction mixture with a volume ratio of 3:1 and was incubated in the boiling water for 5 min. The activity was determined and calculated by the absorbance value at 415 nm. A standard curve of the fucose was made to calculate the amounts of the reducing sugar. The enzyme activity (U) was defined as the amount of enzyme that generated 1 μmoL reducing sugars per minute.

### 2.6. Characterization of Enzyme

The optimum reaction temperature was measured through incubating the reaction mixtures containing 30 μL of the purified OUC-FaFcn1 and 270 μL fucoidan solutions at different temperatures including 25, 30, 35, 40, 45, 50, 55, 60, and 65 °C for 30 min. The relative activity of the OUC-FaFcn1 at the optimum reaction temperature was regarded as 100%. The thermostability of OUC-FaFcn1 was measured by incubating it in the range of 25–50 °C for 48 h, and then tested the residual activity. The enzyme activity was measured following the procedures that are described above.

Analogously, the reaction mixtures containing 30 μL of the purified OUC-FaFcn1 and 270 μL fucoidan solutions with various pHs ranging from 3.0–10.0 were incubated at the optimum reaction temperature for 30 min to determine the optimum reaction pH. The relative activity of the OUC-FaFcn1 at the optimum reaction pH was defined as 100%. The pH stability of OUC-FaFcn1 was investigated by incubating it at a range of pH 6.0–10.0 for 24 h, and then measured the residual activity. 

To determine the effects of different metal ions and chemicals on the activity of OUC-FaFcn1, it was pre-incubated in various metal ions (Na^+^, Mg^2+^, Ba^2+^, Ca^2+^, Zn^2+^, Mn^2+^, Ni^2+^, Co^2+^, Fe^3+^) and chemicals (SDS and Na_2_EDTA) with the concentration of 10 mM at 4 °C for 30 min. The residual activity was measured following the procedures that are described above. Just as mentioned above, the structure of fucoidan is closely associated with its source. To investigate the potential applications of fucoidanase OUC-FaFcn1 in the preparation of various fucoidan oligosaccharides, the fucoidans substrates from *L. japonica* and sea cucumber, including *I. badionotus*, *A. molpadioidea,* and *A. japonicus* were applied to verify the fucoidan-hydrolyzing ability of OUC-FaFcn1 following the procedures that are described above.

### 2.7. Analysis of Hydrolysis Products

The reaction mixtures containing 30 μL of OUC-FaFcn1 (0.17 mg/mL) and 270 μL fucoidan solution (2 mg/mL) were incubated at 40 °C with various times from 0 to 5 min, 15 min, 30 min, 1 h, 2 h, 12 h, 24 h, and 48 h. The degradation products were analyzed by high performance liquid chromatography (HPLC) that was equipped with refractive index detector (RID) (Agilent, Palo Alto, California, USA). In detail, the samples that were pretreated via centrifugation and filtration through a 0.22 μm membrane were loaded on a Superdex^TM^ peptid 30 Increase 10/300 GL column (GE Healthcare, Pittsburgh, Pennsylvania, USA). The mobile phase was 0.2 M NH_4_HCO_3_ with the flow rate of 0.5 mL/min. At the same time, the enzymolysis products were determined by the Agilent Technologies 6460 Triple Quad LC/MS (Agilent, USA) to analyze the Mw of products by electrospray ionization mass spectrometry (ESI-MS) in a negative mode with ion spray voltage of 4 kV and source temperature of 350 °C. 

## 3. Results and Discussion

### 3.1. Sequence Analysis of Fucoidanase OUC-FaFcn1

*F. algicola* 12,076 can grow in medium with fucoidan as the sole carbon source (data not shown), which suggested that this strain can secrete some enzymes that are involved in fucoidan degradation. Previous studies have shown that the strain type *F. algicola* TC2^T^ [30] could digest the GA-fucoidan extracted *Laminaria japonica.* Recently, several enzymes that are involved in the carrageenan metabolism pathway and α-L-fucosidase with transfucosylation activity have been discovered, verified, and analyzed in *F. algicola* [31,32]. However, the enzymes and genes that are responsible for fucoidan degradation in this fucoidan-digesting strain have not been reported. A novel gene encoding a putative fucoidanase OUC-FaFcn1 (OL739608) was cloned from the genome DNA of *F. algicola* 12076. The open reading frame of the fucoidanase gene has 3045 bp and encodes a protein with 1014 amino acid residues. The predicted molecular mass is about 111.4 kDa and there is a signal peptide consisting of 31 amino acids in its *N*-terminus. Modular architecture analysis revealed the presence of various module types of this putative fucoidanase OUC-FaFcn1 (Figure 1A), including a 415-residue-long *N*-terminal domain, three successive repeated Cadherine-like and Immunoglobulin(Ig)-like fold domains, and a 67-amino acid-long secretion system C-terminal protein domain (Por-Secre-tail), which is similar to most members that are recorded in the GH107 family except for P19DFcnA and P5AFcnA that only contain an *N*-terminal catalytic domain. Among them, the *N*-terminal domain is the catalytic region, the Ig-like fold domains play a huge part in the soluble expression of proteins, and the *C*-terminal Por-Secre-tail is involved in outer membrane sorting and covalent modification. Phylogenetic analysis (Figure 1B) showed that OUC-FaFcn1 belongs to the GH107 family and the protein sequence similarity between OUC-FaFcn1 to fucoidanase MfFcnA is 59.01%. These results proved OUC-FaFcn1 is a novel fucoidanase in the GH107 family. Multi-sequence alignment suggested that OUC-FaFcn1 is highly conservative at the active sites of Asp231 and His298 (Figure 1C), which act as catalytic nucleophile and acid/base residue, respectively [24]. By comparing the protein structure of MfFcnA4 (PDB: 6dlh) and MfFcnA9 (PDB: 6dns), OUC-FaFcn1 shares 62.22% and 44.83% structural identities with these two fucoidanases, respectively. The results of the molecular docking simulation between OUC-FaFcn1 and α-L-fucopyranosyl-2-sulfate shows that Asp231 combines with the substrate by hydrogen bond interactions, and His298 acts as an acid-base catalytic action (Figure 1D), which was consistent with that of the novel fucoidanase Fhf2 [26]. A groove on the surface forms a basic pocket that can accommodate and bind to sulphate groups. Thus, the sulfation pattern of fucoidan may play a major role in its recognition of enzymes. This further demonstrated the feasibility of OUC-FaFcn1 in degrading fucoidan. 

### 3.2. Heterologous Expression and Purification of Recombinant OUC-FaFcn1

The host cells of *E. coli* possessing the vector pET-28a(+)-OUC-FaFcn1 were cultivated in ZYP-5052 medium for successful expression of OUC-FaFcn1. After extraction, separation, and purification, a single and clear band of 110 kDa was observed by SDS-PAGE (Figure 2), which is consistent with the result of the theoretical Mw that was calculated from its protein sequence without signal peptide. The Mw of the purified OUC-FaFcn1 is roughly equal to other fucoidanases in GH107 family, such as MfFcnA (105 kDa), FFA1 (111 kDa), and Fp273 (105 kDa), but except P19DFcnA (45.3 kDa) and P5AFcnA (45.2 kDa). The fucoidan-hydrolyzing activity of OUC-FaFcn1 was measured by the pHBH approach, and its specific activity was 4.11 U/mg. Comparing to crude enzymes from 1 L culture broth, the purification fold and the enzyme recovery yield was 3.29 and 59.32%, respectively (Table 1). Generally, fucoidanases possess relatively low activities and only a little research has described the specific activity of fucoidanase. For example, Tatsuhiko [35] researched the ability of fucoidanase from *Luteolibacter algae* H18 towards deacetylated fucoidan, and its activity was 3.18 U/mg. The activity of fucoidanase FcnA from *W. fucanilytica* CZ1127^T^ against fucoidan that was extracted from sea cucumbers *I. badionotus* was 13.7 U/mg [29]. Whereas the activities of other fucoidanases are mostly determined by qualitative determination of products [23,24,25,26,27]. However, the fucoidanase activity is closely related to the source and structure of the substrate fucoidan.

### 3.3. Characteristics of OUC-FaFcn1

The optimal reaction temperature of OUC-FaFcn1 against fucoidan was 40 °C (Figure 3A), and it still showed high fucoidan-hydrolyzing activity when the reaction temperature was between 25 and 55 °C. Most of the characterized fucoidanases showed their highest activities at the temperatures ranging from 25 to 35 °C. For instance, the fucoidanases FFA2, FWf1, FWf2, and Fhf1 unfolded highest activities at 25–37 °C, 24–35 °C, 24–40 °C, and 37 °C [23,25,27]. It suggests that OUC-FaFcn1 has a wide range of temperature adaptations. The relative activity of the purified OUC-FaFcn1 dropped to zero when it was stored at 35 °C or higher within 8 h, whereas when it was stored at 30 and 25 °C, its half-life period exceeded 8 h (Figure 3B). Thus, OUC-FaFcn1 has a relatively stable enzyme activity at low temperatures and its thermal stability needs to be further boosted. The optimal pH of OUC-FaFcn1 for hydrolyzing fucoidan was pH 9.0 (50 mM Tris-HCl buffer), it also kept high enzyme activity in the pH 7.0 and pH 8.0 buffer, whereas it was extremely sensitive to an acidic environment (Figure 3C). Therefore, the OUC-FaFcn1 has high catalytic activity in neutral and weakly alkaline environment which was consistent with the catalytic properties of most fucoidanases except FWf2 (pH 6.0–6.8) [27]. After incubating the OUC-FaFcn1 at the range of pH 6.0–9.0 for 24 h, its residual enzyme activity could exceed 50% (Figure 3D). When it was stored in strong acid and base conditions, its stability was significantly affected (data not shown). To sum up, OUC-FaFcn1 has relatively mild reaction conditions and wide temperature adaptability.

The effects of a serious of metal ions and chemicals on fucoidan-hydrolyzing activity of OUC-FaFcn1 were shown in Figure 3E. Obviously, its fucoidan-hydrolyzing activity was inhibited by Na^+^, Ba^2+^, Ca^2+^, Mn^2+^, Ni^2+^, Fe^3+^, SDS, and Na_2_EDTA, and was stimulated by Mg^2+^, meanwhile, Zn^2+^ and Co^2+^ had no significant effects on fucoidan-hydrolyzing activity. Similarly, some metal ions including Mg^2+^ are important and even essential for the fucoidan-hydrolyzing activities of some fucoidanases such as FWf1, FWf2 [27], FFA1 [22], and FFA2 [23]. Therefore, to improve the degradation efficiency of fucoidan that was catalyzed by OUC-FaFcn1, appropriate amounts of Mg^2+^ can be added into the reaction system.

The structures of fucoidans vary with their sources. Herein, the substrate specificity of fucoidanase OUC-FaFcn1 towards various fucoidans were detected. From the data in Figure 3F, OUC-FaFcn1 could degrade the fucoidan from *Fucales* genus, whereas it hardly hydrolyzed the fucoidans from *L. japonica*, *I. badionotus*, *A. molpadioidea*, and *A. japonicus* with their backbones linked by α-1,3 glycoside bonds. Therefore, it can be speculated that OUC-FaFcn1 only acts on the α-1,4 glycoside bond of fucoidan because the backbone of *Fucales* genus-derived fucoidan is composed of α-1,3 and α-1,4 alternately linked L-fucopyranose residues. It is worth noting that research studies on substrate specificities of the GH107 family fucoidanases are relatively few. Only a few studies have shown that FFA1, FFA2, MfFcnA, FWf1, and FWf2 could specifically cleave α-1,4 glycoside bond between the α-L-fucose residues which are identical to OUC-FaFcn1. However, what should not be ignored is that the location and number of sulfate groups on fucose residues also varies. Recently, a novel fucoidanase, Fhf2, was discovered from *F. haliotis*, which could hydrolyze α-1,4 fucosyl linkages with C2 sulfations [26]. Therefore, the substrate specificity of OUC-FaFcn1 for sulfation pattern remains to be further investigated.

### 3.4. Hydrolysis Process and Principal Product Analysis

The hydrolysis process of the purified OUC-FaFcn1 against the *Fucales* genus-derived fucoidan was detected by HPLC. The results in Figure 4 showed that OUC-FaFcn1 could degrade fucoidan into oligosaccharides with different MWs. At the initial stage of the reaction, the degradation products were a variety of oligosaccharides instead a particular oligosaccharide accumulation, which indicated that OUC-FaFcn1 is an endolytic fucoidanase to cleave the glycoside bond of fucoidan randomly. Till now, most of fucoidanases in the GH107 family are endotype hydrolases [21,22,23,24,25,26,27,28]. In the process of fucoidan degradation, with the reaction time increased, the accumulated amounts of oligosaccharides products were boosted. Particularly, when the reaction time reached 30 min, some new products were introduced and the product I gradually became the master principal product. The amount of product I reached its maximum within 48 h of reaction, which accounted for 49.4% of all the products and there were less products for other aggregations. Due to the complexity of the fucoidan structure, with the increase of the reaction time and the enzyme amount, OUC-FaFcn1 couldn’t degrade the fucoidan completely and up to 75% of the substrate was degraded, which was basically identical to the degradation efficiency of the fucoidanase FFA1 [22]. It was found that the FFA1 treatment of fucoidan from *Sargassum horneri* resulted in the mixture of fucoidan oligosaccharides and fucoidan fragments with a high molecular weight (HMP) which cannot be degraded with the extension of reaction time because the sulfate group that is located on C4 position of the HMP main chain affect the identification and specificity of FFA1 [22].

Afterwards, the principal product Ⅰ was separated and analyzed by ESI-MS and the results in Figure 4 revealed that the corresponding *m*/*z* of product Ⅰ was 283 as [M + H_2_O-2H]^2−^, which was a disaccharide carrying three sulfate groups (Fuc_2_S_3_). The principal product of OUC-FaFcn1 is distinct from that which is produced by other fucoidanases in the GH107 family. For example, both MfFcnA [21] and FFA2 [23] can degrade fucoidan into the final products of tetrasaccharide and hexasaccharide, tetrasaccharide is the principal final product of FAA1 [22] and FWf1 [27], the principal products of Fhf1 [25] include tetrasaccharide, octasaccharide, and decasaccharide, wheread tetrasaccharide, hexasaccharide, and octasaccharide make up the dominating products of FWf2 [27]. To the best of our knowledge, OUC-FaFcn1 is the only biotechnological tool that is available for disaccharide preparation from fucoidan. Commonly, the difference in the degree of product polymerization is related to the minimum identification unit of the enzyme to its substrate. Therefore, it is speculated that the unique products composition that is dominated by disaccharide of the OUC-FaFcn1 is attributable to its smaller recognition substrate. 

## 4. Conclusions

In conclusion, a novel endo-α-1,4-L-fucoidanase OUC-FaFcn1 of GH107 family originating from *F. algicola* 12,076 was discovered and successfully expressed with the specific fucoidan-hydrolyzing activity of 4.11 U/mg. OUC-FaFcn1 had a wide temperature adaptability ranging from 25 to 55 °C and relatively mild reaction conditions. More importantly, OUC-FaFcn1 could efficiently degrade fucoidan into oligosaccharides dominated by disaccharide. OUC-FaFcn1 is a novel and promising biotechnological tool for targeted preparation of specific polymerization oligo-fucoidan, especially disaccharide. 

## Figures and Tables

**Figure 1 foods-11-01025-f001:**
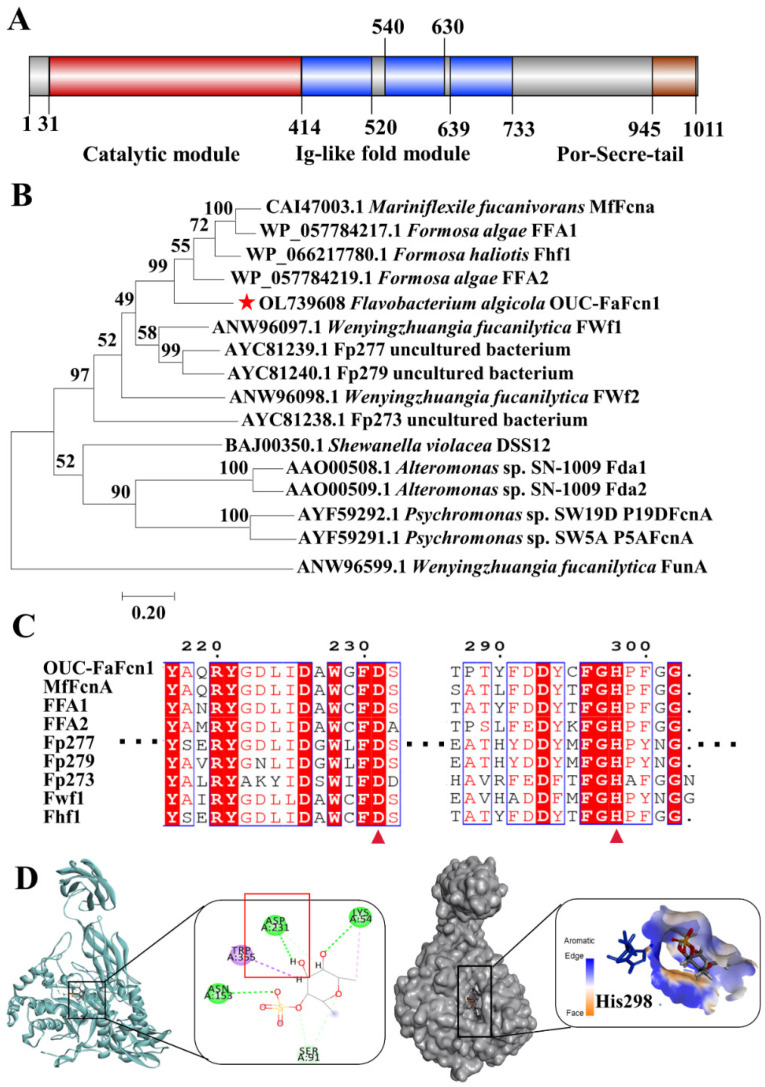
Mining and sequence analysis of α-l-fucoidanase OUC-FaFcn1. (**A**) Protein domain analysis of OUC-FaFcn1. (**B**) Phylogenetic analysis of OUC-FaFcn1 with other α-l-fucoidanases that were recorded in GH107 family. The neighbor-joining tree was obtained using MEGA 7.0. The red star indicates OUC-FaFcn1 of this study. (**C**) Protein sequence alignment of OUC-FaFcn1 with FcnA from *M. fucanivornans* (CAI47003.1), FFA1 and FAA2 from *F. algae* (WP057784217.1, WP 057784219.1), Fhf1 from *F. haliotis* (WP 066217780.1), FWf1 from *W. fucanilytica* CZ1127^T^ (ANW96097.1), Fp273, Fp277, and Fp279 from uncultured bacterium (AYC81238.1, AYC81239.1, AYC81240.1). The two catalytic residues including Asp231 and His298 were marked by a red triangle. (**D**) Homology modeling structure of OUC-FaFcn1. The two catalytic residues including Asp231 and His298 which are located in the catalytic pocket were marked.

**Figure 2 foods-11-01025-f002:**
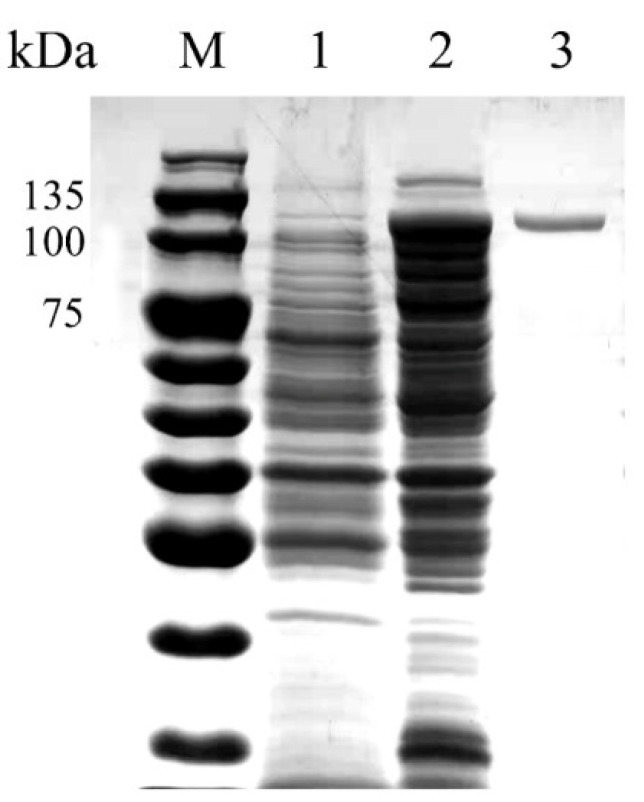
SDS-PAGE analysis of the purified OUC-FaFcn1. Lane M, protein molecular mass marker. Lane 1, the extract of *E. coli* BL21 (DE3) with pET-28a (+) vector (control). Lane 2, the crude enzyme extract. Lane 3, OUC-FaFcn1 purified by Ni-affinity chromatography.

**Figure 3 foods-11-01025-f003:**
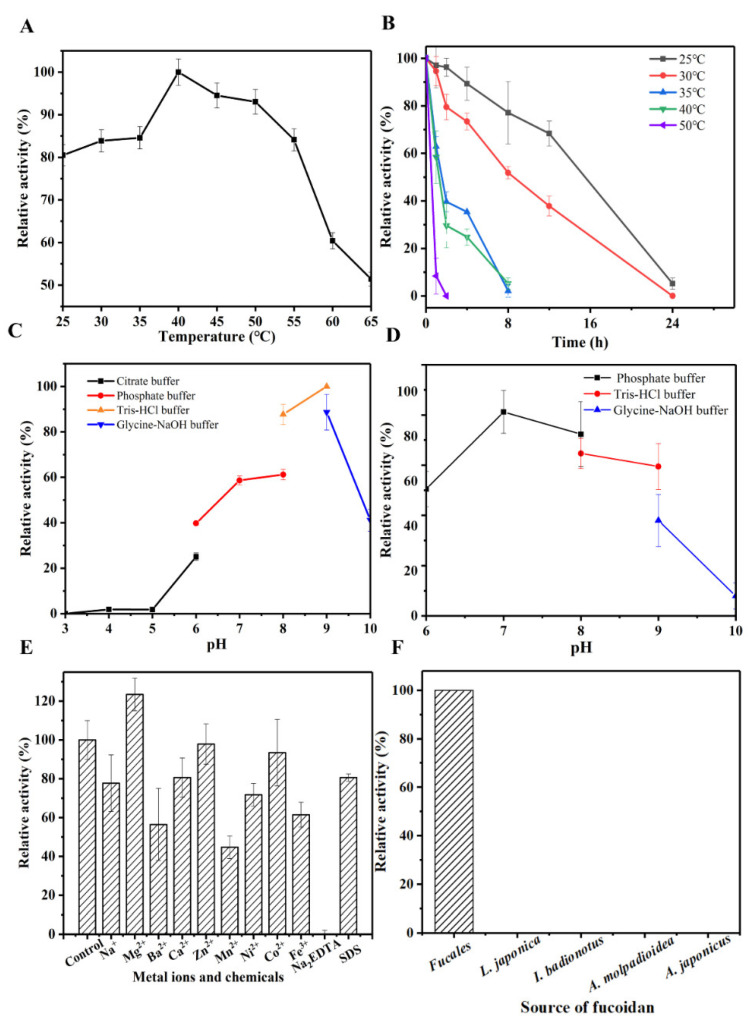
Biochemical characteristics of the purified OUC-FaFcn1. (**A**) Optimum reaction temperature. (**B**) Thermal stability of OUC-FaFcn1. (**C**) Optimum reaction pH. (**D**) pH stability of OUC-FaFcn1. (**E**) Effects of metal ions and chemicals. (**F**) Substrate specificity of OUC-FaFcn1.

**Figure 4 foods-11-01025-f004:**
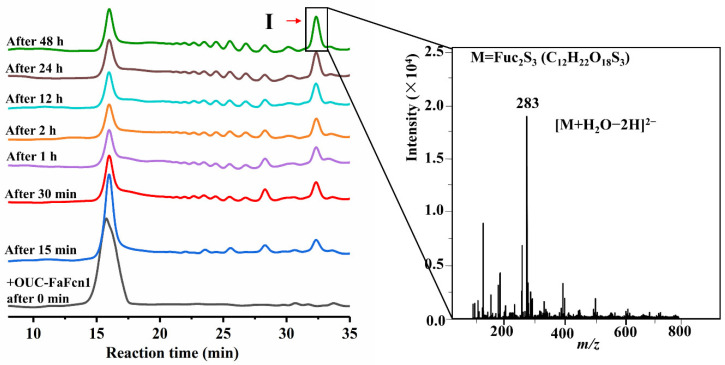
HPLC analysis of degradation products that change over time and the ESI-MS spectrum of the principal product Ⅰ.

**Table 1 foods-11-01025-t001:** Summary of the purification procedures of the recombinant OUC-FaFcn1 that was harvested from 1 L culture broth.

Purification Steps	Total Activity (U)	Total Protein (mg)	Specific Activity (U/mg)	Purification Fold	Yield (%)
Crude enzyme	117.77	94.2	1.25	1	100
Ni-NTA Superflow purification	69.87	17	4.11	3.29	59.32

## Data Availability

Not applicable.

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
