# Peer review of "Expression and Biochemical Characterization of a Novel Fucoidanase from Flavobacteriumalgicola with the Principal Product of Fucoidan-Derived Disaccharide"

_foods, 2022, doi:10.3390/foods11071025_

Round 1

Reviewer 1 Report

     The article is the first to examine the activity of fucoidan-degrading enzyme belonging to GH107 by measuring the reducing end liberated from the substrate, and it is considered to be worthy of evaluation. However, there are some points that need to be revised, so answer the following questions.

  1. It is well known that fucoidan has a different structure depending on the algae. However, this paper examines fucoidan from echinoderms rather than algae in more detail. It seems better to mention the reason.
  2. The result of enzyme purification is mentioned in Table1, but the result from how many liters of culture medium was not shown.
  3. It is written in Fig. 4 that the amount of product 1 is 49.4%, but where does the basis for this value come from?
  4. In Fig. 3, please describe the species names of two algae. And please state the source of fucoidan on line 97.
  5. Line 185 states that the concentration of metal ions and chemicals added is 10 mM, which seems too high in the enzymatic experiments. What do you think?
  6. I think that the specific activity of this enzyme, 4.11 U/mg, should be compared with that previously reported (J. Biosci. Bioeng. 126, 567 (2018)).

Minor points

  1. Please the accession No. (line 117).
  2. There are typographical errors in English (lines 275, 284, and 318).

Reviewer 2 Report

This study is quite interesting and presents some new information on the newly isolated fucoidanase, which has wide adaptability. The manuscript is organized and well-written. The methodology is also suitable. However, a few fixes are required;

  1. The authors should discuss more on the activity of OUC-FaFcn1 in comparison with other available fucoidanase.
  2. The discussion needs to be improved, with more relevant references.
  3. This paper contains a lot of outdated references. Where possible, please update
  4. Check grammar in Line 62, 64 (As a consequent?). The manuscript needs to be checked for grammatical inconsistencies.
